# The impact of screen time and social media on youth self-harm behaviour and suicide: A protocol for a systematic reviews

Kerri M. Gillespie[1,2]*, Grace Branjerdporn[2,3,4,5], Sabine Woerwag Mehta[4,5], Jasmyn Glegg[4], Matthew Porter[2], Selena E. Bartlett[1]

1 School of Clinical Sciences, Faculty of Health, Queensland University of Technology, Kelvin Grove, Queensland, Australia, 2 Mater Research Institute, University of Queensland, St Lucia, Queensland, Australia, 3 Catherine's House for Mothers, Babies and Families, Mater Misericordiae, South Brisbane, Queensland, Australia, 4 Mental Health and Specialist Services, Gold Coast Hospital and Health Service, Southport, Queensland, Australia, 5 Faculty of Health Sciences and Medicine, Bond University, Robina, Queensland, Australia

* k2.gillespie@hdr.qut.edu.au

## Abstract

**Data Availability Statement:** No datasets were generated or analysed during the current study. All

## Introduction

Social media, gaming, and other types of screen time have been associated with a number of child and adolescent mental health concerns, including NSSI, suicidal ideation, suicide attempts and suicides. However, findings have been complicated by a quickly changing technological landscape and the COVID-19 pandemic. Inconsistent findings may be related to the dissimilar impacts of different screen time types on different age groups. The aim of this systematic review is to explore the pattern of impact of different screen time types on children and adolescents, investigating age groups of particular risk, and synthesising outcomes, recommendations, and strategies described to inform future studies and guidelines.

## Methods and analysis

A systematic review will be conducted of all study types, and reported using the Preferred Reporting Items for Systematic review and Meta-Analysis Protocols (PRISMA-P) guidelines. The following databases will be searched for relevant publications: CINAHL, PubMed, Embase, PsycINFO, PsychArticles, Scopus, and Web of Science. Searches will be limited by language (English). Article selection, quality evaluation, and data extraction will be conducted independently by two reviewers. Quality assessment will be conducted using Joanna Briggs Institute (JBI) checklists for the appropriate study type.

## Ethics and dissemination

Ethics approval is not required since we are not using patient data. Findings will be published in a peer-reviewed journal article, and disseminated via conference presentations.

relevant data from this study will be made available upon study completion.

**Funding:** Author SB is partially supported by contributions from the Children's Hospital Research Foundation. No aspect of the research was contingent upon approval by funders. Funders had no involvement in study design, writing of the report, or decision to submit the paper for publication.

**Competing interests:** The authors have declared that no competing interests exist.

## PROSPERO registration number

CRD42023493058.

## Background

Suicide risk and non-suicidal self-injury (NSSI) in young children and adolescents is a growing concern. For more than a decade the rates of suicide and NSSI in young people has been increasing dramatically. A report from the Centres for Disease Control and Prevention (CDC) showed a 57.4% increase in suicides between 2007 and 2018 for young people [1], with suicide currently being the 8[th] leading cause of death in children aged 5–11 in the USA [2]. The COVID-19 pandemic saw even further increases in death by suicides and suicide attempts, particularly in young girls [3, 4]. Globally, it is estimated that approximately 7.5% of children aged 12 years and younger have experienced suicidal ideation, 2.2% have a suicide plan, and 1.3% have attempted suicide [5]. NSSI behaviours are a strong risk factor and predictor for later suicide [6]. Previous studies have estimated the prevalence of NSSI to be approximately 16% in all adolescents, with higher prevalence seen in girls (19.4%) [7].

In line with this increase in child and adolescent NSSI and suicide cognitions and behaviours has been a dramatic rise in the use of online technologies, social media use, and smartphone ownership by teenagers [8]. Current guidelines recommend no screen time for children under two years, one hour per day for children aged two to four, and two hours per day for children aged five and over [9]. However, numerous studies have highlighted that a majority of children from all age groups consistently exceed these recommended timeframes [9–11]. Data compiled since the beginning of the COVID-19 pandemic indicates that 48% of adolescents spend around 5 hours per day on social media, 12% spend more than 10 hours [12], and infants between six and 24 months are exposed to over an hour, and sometimes more than three hours, of screens per day [10].

Well-designed, high-quality content delivered via television or online platforms can be beneficial for children's learning and development [13]. However, evidence from a growing body of research indicates that excessive exposure and inappropriate content may have detrimental effects on language, attention, cognitive development and executive function [14–16]. Screen time has also been linked to increases in anxiety and depressive symptoms, sleep disorders, poor academic performance, impaired social-emotional functioning and inhibitory control issues [17, 18]. In children aged six to 11, increased screen time has been associated with an increase in subsequent suicidal behaviours [19]. While the specific mechanisms behind these relationships is unclear, the displacement of social and tactile engagement and parent-child reading has been a predominant theory [20]. Critical periods of brain development rely on sensory experiences to elicit significant plastic changes [21]. Social interaction is considered necessary for the development of nonverbal communication, language and motor skills, cognitive development, and social development [17]. The replacement of social engagement with screen time can lead to impaired emotion recognition, reduced empathy and curiosity, emotion regulation and behavioural issues, and lower psychological well-being that will likely impact later-life mental health and quality of life [22].

Brain imaging in internet and gaming addiction have observed dopaminergic dysregulation similar to that seen in substance abuse [23]. Internet and gaming addicts were also found to have decreased grey matter volume and structural alterations associated with impaired executive functioning, decision-making, emotional processing, and craving [24]. A systematic

review of young people showed that internet addiction was associated with NSSI, depression and suicidal ideation, while using the internet or gaming for more than five hours per day was associated with suicidal ideations and planning [25]. Unlimited access to online social networks has also changed the landscape of bullying, with cyberbullying becoming a significant threat to child mental health and increasing the risk of NSSI, depression, and suicidal ideation [25]. Increased screen time is also associated with increased sedentary time, reduced physical activity, altered sleep, and depressive symptoms; all of which may further increase the risk of NSSI and suicide cognitions and behaviours [26].

While a growing body of evidence suggests strong correlations between screen time and child mental health, NSSI, and suicidal behaviours, a number of studies have found no relationship [26–29]. This could indicate that the rise in child and adolescent NSSI and suicidal behaviours are associated with alternative societal or environmental factors. However, we contend that these inconsistent findings may be due to the diverse impacts of different types of screen time. Social media in particular has been associated with poor mental health in children [30, 31], while television use has been associated with lower rates of depression and anxiety in some studies [28, 32]. An investigation that compares of all types of screen time is necessary to ensure positive and negative impacts of different types of screen time do not impair the reliability of results.

Despite the growing evidence of the detrimental impacts of screen time on child and adolescent mental health, few systematic reviews have been conducted to investigate these relationships. Prior reviews have primarily addressed impacts on physical health [26] and general psychological wellbeing and neurodevelopment [31, 33, 34], and/or focussed on a single type of screen media (such as social media) [35]. Those that addressed NSSI and suicide have predominantly focussed on cyberbullying [36] or included adults in their sample [25, 37]. This study will investigate and compare the impacts of all facets of screen time to determine the detrimental or protective factors associated with NSSI and suicide cognitions and behaviours, and to identify the ages and demographics of children most at risk.

Parents have previously reported concern and uncertainty regarding screen time and strategies to address the issue [38]. Further education and guidelines regarding these issues could assist parents and help to reduce the risk of mental health associated with screen time. Findings from the current systematic review will be used inform the development of guidelines and recommendations around child screen time for parents, teachers, healthcare professionals, and other professionals who may deal with children at risk of excessive screen time exposure, NSSI, and suicidal cognitions and behaviours.

## Objectives

The objectives of the study are to investigate the impact of screen time on NSSI, suicidal ideation, suicide attempts and suicides in children and adolescents less than 18 years of age. The study will also identify which types of screen time are more detrimental or protective for children. The review will also aim to identify whether different age groups are more vulnerable to the impacts of screen time, and whether different types of screen time are differentially impactful for different age or demographic groups.

## Research questions

1. Is there a relationship between screen time and child and adolescent NSSI, suicide ideation, suicide attempts and death by suicide?

2. Which characteristics or categories of screen time (such as smartphone use, social media platforms, messaging, gaming, television or internet use) are more impactful on child and adolescent NSSI, suicide ideation, suicide attempts and suicides?

3. Are particular age or demographic groups more vulnerable to the detrimental impacts of screen time?

4. What existing interventions or recommendations for screen time use are described in the literature?

## Methods and analysis

This protocol was registered in the International prospective register of systematic reviews (PROSPERO) database with the registration number CRD42023493058, and is reported in accordance with the Preferred Reporting Items for Systematic Review and Meta-Analysis Protocols (PRISMA-P) (see S1 Appendix in Supporting Information for the completed PRISMA-P checklist). The systematic review is anticipated to be conducted over December 2023 to February 2025.

### Outcomes

This review will report on the impacts of different aspects of screen time on children and adolescents. Primary outcomes will include NSSI, suicide ideation, suicide attempts, or lives lost to suicide. Secondary outcomes that may be related to NSSI and suicide cognitions and behaviours will also be investigated, including internet and gaming addiction, mood, and neurodevelopmental impacts. Where possible, findings related to other key variables such as sleep, diet, and sedentary behaviours will be taken into account and discussed for their potential confounding or mediating impact on outcomes.

### Eligibility criteria

**Types of studies.**   This systematic review will include all peer reviewed publications that describe primary data. All qualitative and quantitative study designs will be included. Studies that do not include primary data (review papers, systematic reviews, opinion and commentary papers, editorials, dissertations, posters, and conference abstracts), and secondary analyses of data will be excluded. Should a large number of papers be identified in one or more study design types (such as longitudinal or cross-sectional), final analyses may restrict or compare papers based on this criterion. No location restrictions will be placed on the search. Studies will be limited to those published in 2007 and later. This period begins with the release of the iPhone and will provide a more comparable environment for included studies, in terms of device use and availability. Only studies published in English will be included. The study authors do not have the funding required for the translation of non-English papers. We understand this may lead to omission of some important overseas studies, and this limitation will be discussed in the final manuscript.

**Participants.**   Studies must include children or adolescents aged under 18 years of age.

**Intervention.**   The study must include participant exposure to screen time: which includes use of social media, messaging, video or audio content, or gaming, on a television, smartphone, computer, or other electronic smart device. One or more of these activities must be documented in a study to be included in the final analysis. See Table 1 for further details on inclusion and exclusion criteria.

**Table 1. Inclusion and exclusion criteria.**

| PICOS[a] | Inclusion | Exclusion |
|---|---|---|
| Population | Children and adolescents under 18 years of age. | Adults aged 18 years or older. |
| Intervention/ Exposure | Use of any device that can access internet and streaming services, social media, messaging, or gaming. | NA |
| Comparison | No or fewer hours of screen time. Observational studies may not include a comparison group. | NA |
| Outcomes | NSSI, suicide ideation, suicide attempt, suicide. | Depression, anxiety, or other clinical measure only (without any measure of NSSI, suicide ideation, suicide attempt, or suicide). |
| Study design | Any qualitative or quantitative studies containing primary data (retrospective or prospective cohort studies, cross-sectional studies, randomised control trials) published in English. | Systematic reviews, scoping reviews, literature reviews, dissertations, conference abstracts, posters, letters to the editor, commentary, or opinion papers. Secondary analysis of data. |
| Language | English | Language other than English. |
| Timeframe | Studies published in 2007 to 2024 | Studies published in 2006 or earlier. |
| Setting | No restriction. | NA |

[a]PICOS = Participants, Interventions, Comparators, Outcomes, and Study design.

## Information sources

The systematic review will search the following databases: CINAHL, PubMed, Embase, PsycINFO, PsycArticles, Scopus, and Web of Science. Reference lists of all included papers will also be searched, as will the reference lists of completed systematic reviews relating to child mental health and screen time, to ensure that no eligible papers are overlooked.

## Search strategy

Search terms were chosen after consultation with research team members with expertise in mental health research. Preliminary scoping searches were initially conducted, refining the search strategy to optimise the balance between sensitivity and specificity. The final search will be conducted by a member of the research team. Keywords and MeSH terms relating to the following major concepts were included in the search:

1. Screen time and devices;

2. Commonly used social media platforms and Apps;

3. Age; and

4. NSSI, suicide, and suicidal ideation.

The final review will be conducted using title, abstract and keyword searches. See Fig 1 for the primary search terms used and S2 Appendix in Supporting Information for the full search criteria for each database.

## Study selection

On completion of searches, duplicates will be removed and all identified papers will be imported into Covidence; a web-based software application designed to manage literature and systematic reviews. All screening will be conducted online in Covidence. Due to the large

Screentime OR "Screen time" OR Smartphone OR "smart phone" OR "social media" OR internet OR gaming OR texting OR television OR "video games" OR messaging OR "smart device" OR phone OR telephone OR computer OR laptop OR tablet OR "Chat room" or online or "social network" OR blog OR sext* OR "instant messag*" or "text messag*" OR cyber* OR forum OR YouTube OR reddit OR twitter OR Snapchat OR Instagram OR Facebook OR telegram OR WeChat OR WhatsApp OR TikTok OR discord OR Twitch OR myspace OR Kuaishou OR Qzone OR "Seina Weibo" or QQ OR Threads OR Mastadon OR bluesky OR Tumblr OR Rumble OR Quora

**AND**

Child OR adolescent OR infant OR youth

**AND**

"Self-harm" OR "self harm" OR "self-cut*" OR "self-mutilat*" OR "self-injur*" OR suicide OR suicidal

**Fig 1. Search terms.**

number of papers anticipated in the initial search, title and abstract screening will be conducted by three or more independent reviewers trained in systematic review techniques, with each paper being reviewed by two separate reviewers. Discrepancies between reviewers will be resolved by an independent reviewer who did not participate in title and abstract screening. Papers identified as potentially eligible in the title and abstract screen will be moved to full text screen. Full text screening will be conducted by two independent reviewers, with any discrepancies resolved by a third.

## Risk of bias

Two reviewers will independently assess each study for quality and risk of bias using the appropriate checklist developed by the Joanna Briggs Institute (JBI). Separate versions of the JBI checklist are available for randomised control trials, qualitative, cross-sectional, and cohort studies. These checklists include between eight and 12 items that assess bias associated with sample selection, randomisation, intervention procedures, statistical analysis, and methodological rigour. Items are assessed by answering 'yes (met)', 'no (unmet)', 'unclear' or 'not applicable'. Answers of yes (indicating that a potential bias item was reported and managed appropriately in the study) are divided by the number of all applicable answers to give a quality percentage score. Risk of bias for the study will be determined using the following cutoffs: 70% or more indicates low risk of bias; 50–69% indicates a moderate risk of bias; below 50% indicates a high risk of bias [39]. Should discrepancies between reviewers occur, these will be resolved through discussion. If necessary, a third reviewer will be invited resolve any discrepancy.

## Data extraction

Two independent reviewers will extract data from final papers using pre-defined criteria outlined in a data extraction spreadsheet. Should discrepancies arise in the data extracted, the reviewers will compare results, re-check articles, and discuss findings until a consensus is reached. The following data will be extracted: article authors, publication year, type of study, study date, country, setting, sample size, participant recruitment method, participant characteristics (age, sex, ethnicity), types of screen time, definitions and measures of outcomes,

results, follow-up, attrition, missing data and how it was managed. Where data is missing from an article, authors of the articles will be contacted with a request for these missing data.

## Data synthesis

The main outcomes to be analysed will be prevalence of NSSI, suicidal ideation, suicide attempts, and lives lost to suicide in people under 18 years of age. These outcomes will be assessed for their relationship to screen time; specifically, time exposed to television, gaming, social media, messaging, blogs, or general internet use. Should a minimum of three articles provide sufficient data, we will perform a random-effects meta-analysis. Complete case analysis will be conducted, and the potential implications and limitations of any missing data will be discussed in the final manuscript. Each measure or tool relating to NSSI, suicidal ideation, suicide attempts, and suicides, will be considered separately. We anticipate a large degree of variation between studies in terms of statistical effect measures. However, considering the outcomes will be harm (NSSI or suicide behaviours) or no harm, analyses will most likely be conducted using odds ratios (OR) to determine the odds of self-injurious cognitions and behaviours associated with screen time. Should the majority of papers include continuous outcome variables, the standard mean difference (SMD) will be calculated using means and standard deviations from control and experimental groups. Pooled OR or SMD with 95% confidence intervals and p values will be collated and reported in a forest plot. The degree of heterogeneity will be assessed using the $I^2$ statistic. Sensitivity analysis will be conducted to identify articles that may have disproportionate impacts on the outcomes. Publication bias will be determined using Egger's test.

In the event there is too much variation in statistical measures, or papers provide inappropriate data, we will conduct a narrative synthesis without meta-analysis to synthesise and analyse the relationships in included articles. Where possible, subgroup analyses will be conducted. These will include comparing age groups, sex, screen time use (active versus passive use); type (video or messaging apps, chat rooms, websites, gaming); modality (smartphone, computer, television). A comparison of screen time behaviours before and after COVID-19 will also be conducted and discussed. We anticipate that data will be collected using a mixture of self-report, parent-report, and computer recorded methods. Only data using comparable collection tools and methods will be included in meta-analyses. Random-effects meta-analyses will be used to account for minor heterogeneity.

## Discussion

This study will provide an updated understanding of the impacts of different forms of screen time of child and adolescent NSSI and suicide cognitions and behaviours. It has been proposed that many online activities and practices may be protective against poor mental health, while others are correlated with an increase in NSSI and suicidal ideation, suicide attempts, and death by suicide. We will therefore aim to investigate aspects of screen time and media use independently, in order to determine which aspects of screen time are detrimental and which cohorts may be most at risk. Current recommendations regarding time spent on screens have been criticised for being unachievable or out of touch with our current screen-dominated culture. The study will therefore also examine time spent using screens in order to identify possible dose-response relationships or 'safe' time thresholds in relation to child mental health. Based on our findings, we aim to develop recommendations for screen time use, as well as 'safe' amounts of screen time. Results of this review will inform the funding and development of future studies investigating the impacts of screen time on child mental health. Findings will also guide the development of strategies and guidelines for parents, teachers, and healthcare

professionals, who may come into contact with children who experience detrimental screen use and are at risk of developing injurious behaviours, or the sequalae that precede them.

## Strengths and limitations

Previous systematic reviews have been conducted that investigate screen time, however many include adults in the sample, and do not capture the most recent increase in social media use that is considered to pose a significant risk to mental health. Previous studies have also predominantly focussed on one aspect of screen time. The study will include all screen time in order to control for and differentiate different types of screen time to determine harmful or protective effects. A major limitation of the study will be the exclusion of studies published in languages other than English. However, we believe the findings will capture enough articles to present a representative view of real-world outcomes.

## Supporting information

**S1 Appendix. PRISMA-P 2015 checklist.**
(DOC)

**S2 Appendix. Search strategy.**
(DOCX)

## Author Contributions

**Conceptualization:** Kerri M. Gillespie, Selena E. Bartlett.

**Formal analysis:** Kerri M. Gillespie, Jasmyn Glegg, Matthew Porter.

**Methodology:** Kerri M. Gillespie, Grace Branjerdporn, Sabine Woerwag Mehta, Selena E. Bartlett.

**Writing – original draft:** Kerri M. Gillespie.

**Writing – review & editing:** Grace Branjerdporn, Sabine Woerwag Mehta, Jasmyn Glegg, Matthew Porter, Selena E. Bartlett.

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
