## [Decision Letter · Decision Letter 0]

25 Jun 2024

PONE-D-24-00966The impact of screen time and social media on child NSSI and suicide: a protocol for a systematic reviewPLOS ONE

Dear Dr. Gillespie,

Thank you for submitting your manuscript to PLOS ONE. After careful consideration, we feel that it has merit but does not fully meet PLOS ONE’s publication criteria as it currently stands. Therefore, we invite you to submit a revised version of the manuscript that addresses the points raised during the review process.

We look forward to receiving your revised manuscript.

Kind regards,

Syed Hassan Ahmed

Guest Editor

PLOS ONE

Reviewers' comments:

Reviewer's Responses to Questions

**Comments to the Author**

1. Does the manuscript provide a valid rationale for the proposed study, with clearly identified and justified research questions?

Reviewer #1: Yes

Reviewer #2: Yes

2. Is the protocol technically sound and planned in a manner that will lead to a meaningful outcome and allow testing the stated hypotheses?

Reviewer #1: Yes

Reviewer #2: Yes

3. Is the methodology feasible and described in sufficient detail to allow the work to be replicable?

Reviewer #1: Yes

Reviewer #2: Yes

4. Have the authors described where all data underlying the findings will be made available when the study is complete?

Reviewer #1: Yes

Reviewer #2: No

5. Is the manuscript presented in an intelligible fashion and written in standard English?

Reviewer #1: Yes

Reviewer #2: Yes

6. Review Comments to the Author

You may also provide optional suggestions and comments to authors that they might find helpful in planning their study.

Reviewer #1: The study protocol is well written and demonstrates a clear and logical flow. The objectives and aims are explicitly stated, providing a comprehensive overview of the study's purpose and significance. The methodology is robust and adequately detailed, ensuring reproducibility and validity of the study. The chosen methods are appropriate for the research questions and are well-supported by current literature. I recommend that it is accepted as it is.

Reviewer #2: PONE-D-24-00966

The Impact of Screen Time and Social Media on Child NSSI and Suicide: A Protocol for a Systematic Review

The authors propose to systematically review the literature on the effect of screen time in youth with respect to self-harm behavior and suicide. The authors aim to investigate which types of screen time (smartphone, messaging, social media platforms, gaming, television, or Internet) are detrimental or protective, which age groups are more vulnerable, and if different types of screen time are differentially impactful for different age or demographic groups. The authors have identified a gap in published studies regarding the relationship between screen time and youth mental health in that not all types of screen time have been investigated and compared. The authors intend to resolve this knowledge gap with their proposed protocol for a systematic review.

In general, the paper is fairly well done. There are some inconsistencies in the main text that would benefit from good copyediting. The Reference section will need to be carefully checked. I will make a few comments section-by-section.

Title:

Child implies an age range younger than 12 years old, but a study inclusion criterion is children and adolescents less than 18 years old. Also, I suspect that readers may not be familiar with the NSSI abbreviation. The authors might consider:

“The Impact of Screen Time and Social Media on Youth Self-Harm Behavior and Suicide: A Protocol for a Systematic Review”

Background:

The authors explain in good detail the problem of increased self-harm behavior and suicide in youth parallelling the increased use of online technologies by youth. The goal of the study is stated. Further, the anticipated benefit of the systematic review is stated in the last paragraph.

The authors switch from using the phrase “screen time” to the compound word “screentime” in the Background section and throughout the main text. For clarity and consistency, please choose one or the other. (Figure 1: Search terms, is an exception and in Figure 1 authors are correct to use both “screen time” and “screentime.”)

Objectives:

The objectives of the proposed systematic review are clearly stated.

The first paragraph, first sentence, “The objectives of the study are to investigate the impact of screen time on NSSI, suicidal ideation, suicide attempts and completed suicides in children and adolescents less than 18 years of age”…the preferred phrasing is, “…he/she died by suicide,” and not use “committed suicide” or “completed suicide.” Please reword here and throughout the text (see: https://www.camh.ca/-/media/files/words-matter-suicide-language-guide.pdf).

Research Questions:

The Research Questions do mirror the Objectives. Importantly, it is the first instance in which the types of screen time are itemized: smartphone, messaging, social media platforms, gaming, television, or Internet.

Methods and Analysis:

Instead of placing definite dates of conducting the study, simply state, “The systematic review is anticipated to be conducted over (whatever) months.”

Outcomes:

The primary and secondary outcome variables are listed.

Eligibility Criteria:

The first sentence, “This systematic review will include all peer reviewed publications that describe primary data,” a time interval is typically given, from when-to-when. For example (and the authors may choose whatever time interval they wish), “This systematic review will include all peer reviewed publications that describe primary data from 1972, the introduction of Pong by Atari, to 2023,” or, “This systematic review will include all peer reviewed publications within the past 10 years that describe primary data.”

In Table 1, what is PICOS? It is not explained in the main text nor in Table 1. Perhaps the authors could define PICOS as a footnote in Table 1: Participants, Interventions, Comparators, Outcomes of interest.

The keyword pairings in Figure 1 all are appropriate.

Who will conduct the electronic database search? Will it be a professional librarian or a member(s) of the study team?

Study Selection:

It would be helpful for clarity to add a sentence describing COVIDENCE as an Internet-based application to manage literature and systematic reviews.

Risk of Bias:

The explanation of the assessment for bias utilizing a checklist by the Joanna Briggs Institute is instructive.

I did not understand the relevance of reference #33, Nascimento MB, et al. Role of gluteus maximus and medius activation in the lower limb biomechanical control during functional single- leg tasks: A systematic review. Knee. 2023;43: 163-175. Reference #19 in the Nascimento et al. paper refers to the Checklist for Analytical Cross-Sectional Studies from Joanna Briggs Institute…is that what the authors for the present paper mean to refer to? If so, why not simply borrow the reference #19 from the Nascimento et al. paper?

Data Extraction:

The authors do a good job listing those data to be abstracted from each article and how, along with what to do with missing data.

In line 208, “…characteristics (age, gender, ethnicity),…” I am pretty sure the authors mean “sex” and not “gender”…correct?

Will types of screen time (smartphone, messaging, social media platforms, gaming, television, or Internet) be abstracted?

Data Synthesis:

I am not expert enough in statistics to comment, but I do see that the authors have a plan for a narrative synthesis for data not amenable for formal statistical analysis.

Discussion:

The summary of the proposed systematic review is concisely stated.

References:

Many entries are incomplete. Further, careful attention needs to be given to proper formatting and style, especially what to capitalize, what not to capitalize, and page numbers. Some examples follow:

Reference #1:

Curtin MA. State suicide rates among adolescents and young adults aged 10–24: United States, 2000–2018. Journal Issue. Hyattsville, MD; 2020. Contract No.: 11.

Should be:

Curtin MA. State suicide rates among adolescents and young adults aged 10–24: United States, 2000–2018. Nat Vital Stat Rep. 2020;69: 1-10.

Reference #5:

Geoffroy M-C, Bouchard S, Per M, Khoury B, Chartrand E, Renaud J, et al. Prevalence of Suicidal Ideation and Behaviors in Children Aged 12 Years and Under: A Systematic Review and Meta-Analysis. SSRN Electronic Journal. 2022.

Should be:

Geoffroy M-C, Bouchard S, Per M, Khoury B, Chartrand E, Renaud J, et al. Prevalence of suicidal ideation and behaviors in children aged 12 years and under: A systematic review and meta-analysis. Lancet Psychiatry. 2022;9: 703-714.

Reference #27:

Marciano L, Ostroumova M, Schulz PJ, Camerini AL. Digital Media Use and Adolescents' Mental Health During the Covid-19 Pandemic: A Systematic Review and Meta-Analysis. Front Public Health. 2021;9793868.

Should be (note the need for an appropriately placed colon after the number 9):

Marciano L, Ostroumova M, Schulz PJ, Camerini AL. Digital media use and adolescents' mental health during the COVID-19 pandemic: A systematic review and meta-analysis. Front Public Health. 2021;9: 793868.

7. PLOS authors have the option to publish the peer review history of their article (what does this mean?). If published, this will include your full peer review and any attached files.

Reviewer #1: **Yes: **Thiago Gatti Pianca

Reviewer #2: **Yes: **Alexander M. Scharko, M.D.

---

## [Author Response · Author response to Decision Letter 0]

1 Jul 2024

Please see attached response letter to reviewers, or below responses. 

Title:

Child implies an age range younger than 12 years old, but a study inclusion criterion is children and adolescents less than 18 years old. Also, I suspect that readers may not be familiar with the NSSI abbreviation. The authors might consider:

“The Impact of Screen Time and Social Media on Youth Self-Harm Behavior and Suicide: A Protocol for a Systematic Review”

Response: Thank you for this suggestion. We have amended the title in line with these suggestions.

The authors switch from using the phrase “screen time” to the compound word “screentime” in the Background section and throughout the main text. For clarity and consistency, please choose one or the other. (Figure 1: Search terms, is an exception and in Figure 1 authors are correct to use both “screen time” and “screentime.”)

Response: Thank you for highlighting this. We have now changed all instances of the term to be screen time. 

The first paragraph, first sentence, “The objectives of the study are to investigate the impact of screen time on NSSI, suicidal ideation, suicide attempts and completed suicides in children and adolescents less than 18 years of age”…the preferred phrasing is, “…he/she died by suicide,” and not use “committed suicide” or “completed suicide.” Please reword here and throughout the text (see: https://www.camh.ca/-/media/files/words-matter-suicide-language-guide.pdf).

Response: Thank you for bringing this to my attention. Instances of the phrase committed/completed suicide have been replaced with suicide, death by suicide, died by suicide, or lives lost to suicide, as per the recommended guidelines. 

Methods and Analysis:

Instead of placing definite dates of conducting the study, simply state, “The systematic review is anticipated to be conducted over (whatever) months.”

Response: This has been amended, and dates updated based on current progress, to say:

The systematic review is anticipated to be conducted over December 2023 to August 2024. 

Eligibility Criteria:

The first sentence, “This systematic review will include all peer reviewed publications that describe primary data,” a time interval is typically given, from when-to-when. For example (and the authors may choose whatever time interval they wish), “This systematic review will include all peer reviewed publications that describe primary data from 1972, the introduction of Pong by Atari, to 2023,” or, “This systematic review will include all peer reviewed publications within the past 10 years that describe primary data.”

Response: We have chosen not to restrict the search by date. Line 145 states “no date or location restrictions will be placed on the search”.

In Table 1, what is PICOS? It is not explained in the main text nor in Table 1. Perhaps the authors could define PICOS as a footnote in Table 1: Participants, Interventions, Comparators, Outcomes of interest.

Response: We have now included what PICOS stands for in a footnote under the first table. 

Who will conduct the electronic database search? Will it be a professional librarian or a member(s) of the study team?

Response: The search will be conducted by a member of the research team after consultation with experts and a preliminary, refining search (see lines 1666-169 on page 9).

Study Selection:

It would be helpful for clarity to add a sentence describing COVIDENCE as an Internet-based application to manage literature and systematic reviews.

Response: This additional description of Covidence has now been included in lines 183-185 of page 10. 

I did not understand the relevance of reference #33, Nascimento MB, et al. Role of gluteus maximus and medius activation in the lower limb biomechanical control during functional single- leg tasks: A systematic review. Knee. 2023;43: 163-175. Reference #19 in the Nascimento et al. paper refers to the Checklist for Analytical Cross-Sectional Studies from Joanna Briggs Institute…is that what the authors for the present paper mean to refer to? If so, why not simply borrow the reference #19 from the Nascimento et al. paper?

Response: I see how the inclusion of this reference seems out of place. However, we have included that reference for their use of the specific cut-offs that we are including in our review. The recommended JBI reference #19 refers to the checklist, but does not suggest a cut-off for the interpretation of the checklist. 

In line 208, “…characteristics (age, gender, ethnicity),…” I am pretty sure the authors mean “sex” and not “gender”…correct?

Response: We did intend this to be sex and not gender. This has now been amended. 

Will types of screen time (smartphone, messaging, social media platforms, gaming, television, or Internet) be abstracted?

Response: Thank you for highlighting this. We have now included “types of screen time” in the list of data to be extracted from papers (line 212 on page 11). 

References:

Many entries are incomplete. Further, careful attention needs to be given to proper formatting and style, especially what to capitalize, what not to capitalize, and page numbers. Some examples follow:

Response: Thank you for highlighting these errors. We have now re-edited the reference section and amended any errors identified. 

Additional changes

Response: Please note that due to changing roles within the research team, and after reviewing work conducted on this project by researchers, I have removed the author Xin Li and replaced with author Matthew Porter.

---

## [Decision Letter · Decision Letter 1]

22 Oct 2024

PONE-D-24-00966R1The impact of screen time and social media on youth self-harm behaviour and suicide: A protocol for a systematic reviewPLOS ONE

Dear Dr. Gillespie,

Thank you for submitting your manuscript to PLOS ONE. After careful consideration, we feel that it has merit but does not fully meet PLOS ONE’s publication criteria as it currently stands. Therefore, we invite you to submit a revised version of the manuscript that addresses the points raised during the review process.

We look forward to receiving your revised manuscript.

Kind regards,

Monika Sreeja Thangada, M.D.

Guest Editor

PLOS ONE

Journal Requirements:

Reviewers' comments:

Reviewer's Responses to Questions

**Comments to the Author**

1. Does the manuscript provide a valid rationale for the proposed study, with clearly identified and justified research questions?

Reviewer #1: Yes

Reviewer #2: Yes

Reviewer #3: Yes

Reviewer #4: Yes

2. Is the protocol technically sound and planned in a manner that will lead to a meaningful outcome and allow testing the stated hypotheses?

Reviewer #1: Yes

Reviewer #2: Yes

Reviewer #3: Yes

Reviewer #4: Yes

3. Is the methodology feasible and described in sufficient detail to allow the work to be replicable?

Reviewer #1: Yes

Reviewer #2: Yes

Reviewer #3: Yes

Reviewer #4: Yes

4. Have the authors described where all data underlying the findings will be made available when the study is complete?

Reviewer #1: Yes

Reviewer #2: Yes

Reviewer #3: Yes

Reviewer #4: Yes

5. Is the manuscript presented in an intelligible fashion and written in standard English?

Reviewer #1: Yes

Reviewer #2: Yes

Reviewer #3: Yes

Reviewer #4: Yes

6. Review Comments to the Author

You may also provide optional suggestions and comments to authors that they might find helpful in planning their study.

Reviewer #1: The paper was good enough in the last version, and I have nothing else to add to it at this point in time.

Reviewer #2: The Impact of Screen Time and Social Media on Youth Self-Harm Behaviour and Suicide: A Protocol for a Systematic Review

PONE-D-24-00966R1

The original manuscript was fairly well written and this revision has improved the manuscript. I only have two points for feedback.

1. On page 6 of the revised manuscript, Methods and Analysis, opening paragraph indicates that the review is anticipated to be conducted over December 2023 to August 2024, but we are already in September 2024…has the review already been accomplished?

2. The Reference section is better, but still needs some work. Please make sure the references are properly formatted.

Reviewer #3: Thank you for the opportunity to review this manuscript. The study provides valuable insights into the relationship between screen time and mental health outcomes in children and adolescents, addressing an important topic in today's digital age. The manuscript is well-structured and presents its methodology clearly. Below are my comments.

Introduction: Consider differentiating between active screen time (e.g., social media, gaming) and passive screen time (e.g., TV watching), as evidence suggests they may have different effects on mental health.

Children and adolescents are in critical stages of brain development, and prolonged screen time may affect neural pathways related to attention and emotional regulation. Discuss how screen time could impact this. Also, consider potential long-term mental health effects.

Include a discussion on how screen time, particularly from social media or gaming, triggers dopamine release, leading to addictive behaviors that may contribute to mental health problems like anxiety, depression, or NSSI.

Methods : The need for subgroup analysis is mentioned, but no criteria is provided. Specify which subgroups will be analyzed (e.g., by age, type of screen time).

Explain how studies with varying definitions of screen time (e.g., self-reported vs. objectively measured) will be standardized and integrated into the analysis.

The manuscript mentions contacting authors for missing data but does not provide further steps if no response is obtained. Consider adding alternative strategies such as data imputation or sensitivity analyses.

Discussion : Discuss whether any conclusions can be drawn regarding specific thresholds or “safe” amounts of screen time, particularly in relation to mental health outcomes.

Reviewer #4: The study "The Impact of Screen Time and Social Media on Youth Self-Harm Behaviour and Suicide" outlines a systematic review to assess the relationship between screen time, non-suicidal self-injury (NSSI), suicidal ideation, and suicide among children and adolescents.

1. The study does not specify any time limitations on the studies to be included, which could result in the inclusion of outdated research. Given that technology and digital behaviors have changed significantly over the past decade, including studies from vastly different time periods might complicate synthesis and lead to misleading conclusions.

2. The study acknowledges that the pandemic has affected screen time and mental health behaviors, yet it does not indicate how studies conducted before and during the pandemic will be compared. Screen time behaviors and their psychological effects may have shifted dramatically during COVID-19, necessitating a more explicit plan for handling this potential confounder.

3. While the study rightly aims to examine different types of screen time, such as social media, gaming, and television, the inclusion of vastly different types of media in a single analysis might lead to significant heterogeneity. The outcomes associated with social media use, for example, may be very different from those associated with passive television watching, which could weaken the study's conclusions

7. PLOS authors have the option to publish the peer review history of their article (what does this mean?). If published, this will include your full peer review and any attached files.

Reviewer #1: No

Reviewer #2: **Yes: **Alexander M. Scharko, M.D.

Reviewer #3: **Yes: **Mohsin Raza

Reviewer #4: **Yes: **Nikhil Tondehal

---

## [Author Response · Author response to Decision Letter 1]

3 Nov 2024

1. On page 6 of the revised manuscript, Methods and Analysis, opening paragraph indicates that the review is anticipated to be conducted over December 2023 to August 2024, but we are already in September 2024…has the review already been accomplished?

Response: Unfortunately, our progress has been delayed due to a series of illnesses, deaths, and births in the families of all of the primary reviewers. We have updated the timeframes in the manuscript for this, and plan to conduct an updated search of the intervening time period using the same methods outlined, before analysis is conducted. We now estimate that the review will be completed by February 2025.

2. The Reference section is better, but still needs some work. Please make sure the references are properly formatted.

Response: Thank you for highlighting this. We have now formatted the reference section.

3. Introduction: Consider differentiating between active screen time (e.g., social media, gaming) and passive screen time (e.g., TV watching), as evidence suggests they may have different effects on mental health.

Response: We have now included this in the section detailing subgroup analysis. This will be a very important aspect that we will investigate. 

4. Children and adolescents are in critical stages of brain development, and prolonged screen time may affect neural pathways related to attention and emotional regulation. Discuss how screen time could impact this. Also, consider potential long-term mental health effects.

Response: Additional text has been included in the background (page 4) that discusses potential mechanisms and impacts on later life mental health. 

5. Include a discussion on how screen time, particularly from social media or gaming, triggers dopamine release, leading to addictive behaviors that may contribute to mental health problems like anxiety, depression, or NSSI.

Response: The addictive nature of internet use (including gaming and social media), and the neurobiological similarities to substance use, has been described in more detail on page 4. 

6. Methods: The need for subgroup analysis is mentioned, but no criteria is provided. Specify which subgroups will be analyzed (e.g., by age, type of screen time). Explain how studies with varying definitions of screen time (e.g., self-reported vs. objectively measured) will be standardized and integrated into the analysis.

Response: Thank you for highlighting this. We have included further information regarding the subgroups we will analyse. We have also described how different data collection methods will be dealt with on page 13:

"These will include comparing age groups, sex, screen time use (active versus passive use); type (video or messaging apps, chat rooms, websites, gaming); modality (smartphone, computer, television). A comparison of screen time behaviours before and after COVID-19 will also be conducted and discussed. We anticipate that data will be collected using a mixture of self-report, parent-report, and computer recorded methods. Only data using comparable collection tools and methods will be included in meta-analyses. Random-effects meta-analyses will be used to account for minor heterogeneity."

7. The manuscript mentions contacting authors for missing data but does not provide further steps if no response is obtained. Consider adding alternative strategies such as data imputation or sensitivity analyses.

Response: While we recognise that failing to deal with missing data using statistical methods such as imputation may lead to bias in the results, it is also understood that these statistical methods can significantly alter and potentially bias the outcomes of a meta-analysis (Kahale et al., 2020).

In order to maintain transparency and replicability in a systematic review that may include studies using a range of analytic methods, the study will use a complete case analysis. However, we will discuss the limitations of missing data in the manuscript, and describe how these may potentially contribute to bias in the findings.

This has now been described under ‘Data Synthesis’ on page 12.

Kahale LA, Khamis AM, Diab B, Change Y, Lopes LC. 2020. Potential impact of missing outcome data on treatment effects in systematic reviews: Imputation study. BMJ. 370, m2898.

8. Discussion: Discuss whether any conclusions can be drawn regarding specific thresholds or “safe” amounts of screen time, particularly in relation to mental health outcomes.

Response: The following sentences have been included in the discussion section: 

Current recommendations regarding time spent on screens have been criticised for being unachievable or out of touch with our current screen-dominated culture. The study will therefore also examine time spent using screens in order to identify possible dose-response relationships or ‘safe’ time thresholds in relation to child mental health. Based on our findings, we aim to develop recommendations for screen time use, as well as ‘safe’ amounts of screen time. 

1. The study does not specify any time limitations on the studies to be included, which could result in the inclusion of outdated research. Given that technology and digital behaviors have changed significantly over the past decade, including studies from vastly different time periods might complicate synthesis and lead to misleading conclusions.

Response: Thank you for highlighting this issue. This had been previously discussed with the research team. On further discussion, we have decided to limit the study to papers published in 2007 and later. This is the year that the iPhone was released and widespread use of social media apps began after this time. This will provide a more comparable social and technical environment for included studies. 

2. The study acknowledges that the pandemic has affected screen time and mental health behaviors, yet it does not indicate how studies conducted before and during the pandemic will be compared. Screen time behaviors and their psychological effects may have shifted dramatically during COVID-19, necessitating a more explicit plan for handling this potential confounder.

Response: We have now included this in our description of subgroup analyses. We will need to investigate any differences in screen time use, and in prevalence of self-harm and suicide associated with this screen time, before and after the COVID-19 pandemic. 

3. While the study rightly aims to examine different types of screen time, such as social media, gaming, and television, the inclusion of vastly different types of media in a single analysis might lead to significant heterogeneity. The outcomes associated with social media use, for example, may be very different from those associated with passive television watching, which could weaken the study's conclusions

Response: We understand that there is a risk with including such a wide range of screen time types. However, the primary purpose of the review is to identify the differences between these types of screen time, and the ways they are used. 

A number of researchers have reported benefits of some services, while detrimental impacts of others. It is therefore important we identify the factors that make particular types of screen time beneficial for children as opposed to harmful. 

We will also investigate the differences between passive and active forms of use. This has now been included in our more detailed description of subgroup analyses on page 12-13 (as described above).

---

## [Editor Report · Decision Letter 2]

14 Nov 2024

The impact of screen time and social media on youth self-harm behaviour and suicide: A protocol for a systematic review

PONE-D-24-00966R2

Dear Dr. Gillespie,

We’re pleased to inform you that your manuscript has been judged scientifically suitable for publication and will be formally accepted for publication once it meets all outstanding technical requirements.

Kind regards,

Monika Sreeja Thangada, M.D.

Guest Editor

PLOS ONE
---

## [Editor Report · Acceptance letter]

18 Nov 2024

PONE-D-24-00966R2 

PLOS ONE

Dear Dr. Gillespie, 

I'm pleased to inform you that your manuscript has been deemed suitable for publication in PLOS ONE. Congratulations! Your manuscript is now being handed over to our production team.

Kind regards, 

on behalf of

Dr. Monika Sreeja Thangada 

Guest Editor

PLOS ONE